# A–Z of Epigenetic Readers: Targeting Alternative Splicing and Histone Modification Variants in Cancer

**DOI:** 10.3390/cancers16061104

**Published:** 2024-03-09

**Authors:** Nivedhitha Mohan, Roderick H. Dashwood, Praveen Rajendran

**Affiliations:** 1Center for Epigenetics & Disease Prevention, Texas A&M Health, Houston, TX 77030, USA; 2Department of Translational Medical Sciences, Antibody & Biopharmaceuticals Core, Texas A&M School of Medicine, Houston, TX 77030, USA

**Keywords:** epigenetic reader, alternative splicing, precision oncology, isoform

## Abstract

**Simple Summary:**

Alternative splicing is a pivotal regulatory mechanism in gene expression, generating multiple RNA species and influencing protein isoform diversity and function, as well as regulatory RNA networks. Histone post-translational modifications can affect alternative splicing and mediate diverse interactions between histones and splicing factors, adding complexity to the transcriptional landscape. Alternative splicing of epigenetic ‘reader’ proteins is of growing interest in cancer etiology and other pathologies, providing insights into effective new therapeutic strategies, including RNA-based approaches and small-molecule targeting of splicing abnormalities.

**Abstract:**

Epigenetic ‘reader’ proteins, which have evolved to interact with specific chromatin modifications, play pivotal roles in gene regulation. There is growing interest in the alternative splicing mechanisms that affect the functionality of such epigenetic readers in cancer etiology. The current review considers how deregulation of epigenetic processes and alternative splicing events contribute to pathophysiology. An A–Z guide of epigenetic readers is provided, delineating the antagonistic ‘yin-yang’ roles of full-length versus spliced isoforms, where this is known from the literature. The examples discussed underscore the key contributions of epigenetic readers in transcriptional regulation, early development, and cancer. Clinical implications are considered, offering insights into precision oncology and targeted therapies focused on epigenetic readers that have undergone alternative splicing events during disease pathogenesis. This review underscores the fundamental importance of alternative splicing events in the context of epigenetic readers while emphasizing the critical need for improved understanding of functional diversity, regulatory mechanisms, and future therapeutic potential.

## 1. Introduction

Eukaryotic DNA is organized into chromatin, encompassing the histone protein core around which DNA is wrapped. Chromatin switches dynamically between closed and open configurations depending on the state of repressed versus activated transcription, respectively [1,2]. Post-translational modification (PTM) of the core histone proteins plays a key role in the regulation of chromatin in its various active states [2]. A variety of PTMs, including acetylation, methylation, phosphorylation, ubiquitination, and SUMOylation, occur in the histone tails, thereby controlling chromatin accessibility [1]. 

Histone ‘reader’, ‘writer’, and ‘eraser’ proteins recognize, add, or remove the PTMs in chromatin and have critical roles in gene regulation during physiological and pathophysiological processes [3]. Recent attention has been directed towards the inhibition of epigenetic reader proteins with anticancer agents that target specific histone acetylation marks in chromatin, effectively downregulating oncogenic targets like MYC [4,5]. The realization that epigenetic readers also undergo alternative splicing events that affect functionality and therapeutic efficacy is an evolving area of research interest [6]. 

Alternative splicing is the process of editing nascent or precursor messenger mRNA (pre-mRNA) to mature mRNA. Editing involves the excision of introns and the ligation of exons through a cascade of reactions facilitated by the spliceosome complex. Splicing can occur co-transcriptionally, and this connection between RNA splicing and concurrent transcription suggests that DNA-associated elements that impact transcriptional regulation also influence splicing events [7]. Numerous reports support the functional coupling between transcription and splicing [7,8,9,10,11], first suggested over three decades ago based on the observation that the spacing between consecutive splice site junctions exhibited a periodic pattern, aligning with the phasing of nucleosomes [8]. Technological advancements in genomics, including genome-wide tiling arrays and sequencing, generated detailed nucleosome and chromatin state maps, revealing enrichment of histone H3 Lys36 trimethylation (H3K36me3) within exons compared to introns, underscoring the connection between nucleosome positioning and exon structure [9,10]. Chromatin architecture can also impact alternative splicing, as evidenced in a study in which alternative exons exhibited weaker nucleosome peaks than constitutive exons in exon arrays [11]. Moreover, the incorporation levels of alternative exons could be predicted based on their respective nucleosome occupancy [11]. 

Alternative splicing is an important regulatory mechanism that facilitates the production of multiple mRNA variants from a single gene [12]. Similar mechanisms apply to non-coding RNAs and regulatory RNA networks, which are not the subject of the current review. Alternatively spliced mRNA encodes distinct protein variants that influence various cellular processes, including the maintenance of homeostasis as well as tissue specificity and disease development [12]. Alternative splicing employs five different mechanisms to orchestrate the network of gene regulation, namely (i) exon skipping, (ii) alternative 5′-donor sites (A5’SS), (iii) alternative 3′-acceptor sites (A3’SS), (iv) mutually exclusive exons, and (v) intron retention. These events can influence mRNA stability, sub-cellular localization, protein translation, and alter the reading frame, resulting in the generation of isoforms with diverse functions [13]. Remarkably, prior reports suggested that 90–95% of human genes undergo alternative splicing [14,15]. High-resolution mass spectrometry revealed that out of ∼20,000 human protein-coding genes, almost 37% generate multiple protein isoforms, thereby contributing to proteome complexity [16]. 

The expansion of epigenomic technologies and next-generation sequencing provided new insights into alternative splicing, including epigenetic reader variants, adding another layer of regulatory complexity to the transcriptional landscape. Acetylation, methylation, and phosphorylation are major histone PTMs that affect alternative splicing or mediate the interaction of histones with splicing factors to drive signal transduction [1,17,18]. Less prominent chromatin marks, such as histone butyrylation and the corresponding butyryl readers, as well as RNA-binding proteins, have also been implicated in alternative splicing. We obtained documented evidence for 13 acetyl readers, 14 methyl readers, 2 phosphoryl readers, 3 butyryl readers, and 6 RNA-binding readers, which either undergo or regulate the alternative splicing of related gene transcripts (Figure 1a) and involve mechanisms (i)–(v) mentioned above (Figure 1b,c). These aspects will be discussed in further detail, along with consideration of alternative splicing in cancer etiology and potential impacts on drug resistance and radiosensitization. 

## 2. Alternative Splicing of Acetyl Readers

### 2.1. Acetyl Readers in Cancer

The rational design of JQ1 as a competitive inhibitor of Bromodomain Containing Protein 4 (BRD4) [19] opened the avenue for molecularly-targeted therapeutic agents geared towards epigenetic reader proteins with key roles in homeostasis and cancer [20]. BRD4 and other members of the Bromodomain and Extraterminal Domain (BET) family bind acetylated histones and play a crucial role in gene transcription [20]. Interestingly, BRD4 depletion led to changes in splice patterns, particularly exon skipping, and co-localization with the splicing regulator Fused in Sarcoma (FUS) on chromatin [21]. BRD4-depleted cells subjected to heat stress showed increased splicing inhibition, especially intron retention, resulting in reduced mRNA levels. BRD4 also interacted with heat shock factor 1 (HSF1) recruited to nuclear stress bodies, highlighting its role in responding to cellular heat stress [22]. 

NUT carcinoma, a rare but highly malignant cancer, is caused by genomic rearrangements involving the *BRD4* or *BRD3* and Nuclear Protein in Testis (*NUT*) genes [20]. For example, the oncogenic transcript *BRD4-NUT* ex15: ex2nt1-585 is formed by the fusion of *BRD4* exon 15 with *NUT* exon 2. This fusion occurs due to the disruption of the canonical 3’ acceptor splice site in *NUT* gene intron 1, which induces alternative splicing from the cryptic splice site [23]. 

Highlighting the potential ‘yin-yang’ nature of alternative splicing, the short isoform of BRD4 (BRD4-S) generated from exon 12–20 skipping was oncogenic in breast cancer cell proliferation and metastasis, whereas the long isoform BRD4-L exhibited tumor suppressor properties [24,25]. In triple-negative breast cancer (TNBC), BRD4-S was co-regulated with the transcription factor Engrailed-1 (EN1) and influenced extracellular matrix-associated cancer-related genes and pathways [24,25]. Two isoforms of BRD3 have also been reported, which display differential binding affinities towards acetylated histones involved in the regulation of mitosis [26]. 

A member of the Switch/Sucrose Non-Fermentable (SWI/SNF) chromatin remodeling complex, SWI/SNF-related matrix-associated actin-dependent regulator of chromatin, subfamily a, member 2 (SMARCA2), through alternative splicing gives rise to *SMARCA2a* and *SMARCA2b* major transcripts, plus multiple additional minor variants. SMARCA2b exhibits higher expression in certain cell lines, which are more sensitive to serum starvation than cells expressing SMARCA2a. Cyclin D1 regulates both SMARCA2a and SMARCA2b, suggesting that SMARCA2 may have distinct functions based on its splice forms, depending on differences in abundance [27]. 

SMARCA4, another member of the SWI/SNF family, undergoes alternative splicing to produce four distinct isoforms [28]. One SMARCA4 isoform arises from a non-canonical splice site and is found in brain, muscle, and germline tissues but is absent from most common tumor types, including lung cancers [29,30]. This isoform contains a novel exon of 99 bp with no similarity to its paralog, SMARCA2 [31]. Another tissue-specific SMARCA4 isoform includes an additional exon of 96 bp between exons 28 and 29 [32]. These findings highlight the range of SMARCA4 isoforms in different tissues and their potential for divergent mechanisms in human cancer [32].

Another multifaceted protein that acts either as a tumor promoter or suppressor in various tissues is Zinc Finger MYND-Type Containing 8 (ZMYND8), exhibiting diverse effects on cell proliferation, angiogenesis, invasion, and metastasis [33]. As a result of alternative splicing, *ZMYND8* exon 22 inclusion was linked to a better prognosis in breast cancer patients, enhanced by hnRNPM and opposed by Epithelial Splicing Regulatory Protein 1 (ESRP1) [34]. This inclusion event was linked to suppression of epithelial-to-mesenchymal transition (EMT) and genes associated with less hormone-dependent and treatment-resistant subtypes. Thus, *ZMYND8* exon 22 inclusion can play a pivotal role in shaping clinical outcomes for breast cancer patients [34].

Another epigenetic reader protein, p300, enhances genome-wide exon inclusion and impacts alternative splicing in breast cancer cells [35]. p300 associates with the *CD44* promoter, promoting *CD44v* exon inclusion, altering cellular traits by detaching Heterogenous Ribonuclear Protein M (hnRNPM) from CD44 pre-mRNA, and activating SRC Associated in Mitosis of 68 kDa (Sam68), ultimately reducing cell motility and promoting epithelial characteristics. This underscores the role of p300 in chromatin-mediated alternative splicing in breast cancer [35].

ZMYND11, also known as BS69, is a transcriptional repressor with distinctive chromatin-binding domains, selectively recognizing histone variant H3.3 lysine 36 trimethylation (H3.3K36me3). ZMYND11 is linked to U5 snRNP spliceosome components and plays a role in governing intron retention by interacting with Elongation Factor Tu GTP Binding Domain Containing 2 (EFTUD2). This regulation is tied to the ability of EFTUD2 to interact with chromatin marked by H3K36me3 in lung cancer cells. Thus, ZMYND11 is a unique reader of H3.3K36me3, influencing pre-mRNA processing and alternative splicing by regulating intron retention [36].

### 2.2. Acetyl Readers in Other Developmental Processes

In Drosophila, the Nucleosome Remodeling Factor (NURF) complex plays a key role in ATP-dependent nucleosome sliding, influencing gene regulation and chromatin structure. Within the NURF complex, the full-length subunit NURF301 recognizes specific histone modifications and is essential for spermatogenesis, while alternative splice forms exhibit spermatocyte arrest, highlighting the critical role of NURF complexes in spermatocyte differentiation [37]. 

A testes-specific mutation in the *Bromodomain Testis Associated* (*BRDT*) gene (*BRDT BD1*) leads to male sterility by impairing sperm development. This also affects over 400 regulatory genes, particularly those involved in RNA splicing, with unusual 3′-UTR features [38]. Alternatively spliced BRDT isoform BRDT-NY was found to be exclusively expressed in higher amounts in adult testis tissues, suggesting its essential roles in spermatogenesis and male sterility [39]. 

Alternative splicing of TATA-Box Binding Protein Associated Protein 1 (TAF1) fine-tunes transcription in the reaction to DNA damage and developmental cues [40]. Genome-wide studies indicated that Bromodomain-containing protein 2 (BRD2) modulated the alternative splicing patterns of 290 genes among the 1450 genes regulated by BRD2 [41]. Alternatively spliced BRD2 exhibits tissue-specific expression in different neural tissues and is also linked to electroencephalographic abnormalities and epilepsy in humans [42]. Specific *Cat eye syndrome chromosome region candidate 2* (*Cecr2*) splice variants in mice cause neural tube defects, underscoring the involvement of CECR2 in neurulation [43]. Bromodomain and PHD finger-containing protein 1 (BRPF1) possesses multiple isoforms and regulates gene expression, DNA repair, and stem cells [44]. BRPF1B binds to histones and is sensitive to inhibitors, while BRPF1A cannot bind due to a specific insertion associated with alternative splicing. Targeting BRPF1B therapeutically can potentially treat bone-related diseases by controlling excessive osteoclast formation [44]. More details on the effects of alternative splicing events on various acetylation readers are compiled in Table 1.

## 3. Alternative Splicing of Methylation Readers 

### 3.1. Methyl Readers in Cancer

Krϋppel-like factor 4 (KLF4) is a transcription factor with roles in stemness, embryonic development, and cancer, acting either as an oncogene or tumor suppressor [45]. A splice variant, *KLF4α*, produced a truncated protein lacking key functional domains. In melanoma, variable expression of KLF4α may enhance pro-tumorigenic roles [45]. In breast cancer, the KLF4α/KLF4 (full length) (FL) ratio increased in tumors, leading to reduced expression of KLF4-L target genes and increased cell proliferation. By interacting with and sequestering KLF4-L in the cytoplasm, KLF4α or KLF4-S counteracted critical nuclear regulatory functions, thereby adding complexity to the role of KLF4 in breast cancer [46].

Another interesting example of alternative splicing involves *Wilm’s Tumor 1* (*WT1*) gene products. Specifically, the WT1-S isoform accelerated tumor growth, while WT1-L suppressed tumor characteristics. However, outcomes were also dependent on the cell line; the WT1-L isoform suppressed tumorigenicity in 7C3H2 but not in 7C1T1 cells [47]. Introducing transcription factor Early Growth Response Factor 1 (EGR1) into 7C3H2 cells enhanced growth and counteracted the tumor-suppressing role of the WT1-L isoform, emphasizing the significance of upstream regulators of alternative spliced isoforms in disease models [47]. 

Ubiquitin Like with PHD and Ring Finger Domains 2 (UHRF2), is a cell cycle regulator that promotes cell proliferation and antagonizes tumor suppressor gene expression in breast cancer [48]. Bioinformatic reports indicated that increased incorporation of UHRF2 exon 10 is associated with normal samples [49]. This inclusion may disrupt the UHRF2 SRA-YDG domain, which is essential for binding to epigenetic marks. Consequently, alternative splicing might curb UHRF2 oncogenic roles in breast cancer through the generation of a truncated protein. Notably, UHRF2 total expression remained consistent between tumor stage I and normal samples, bolstering the validity of the bioinformatic observation as it does not depend on total changes in gene expression [49]. 

Methyl-CpG Binding Domain Protein 3 (MBD3) depletion leads to the dysregulation of Breast Cancer Gene 1 (*BRCA1*) alternative splicing via a long non-coding RNA in glioblastoma cells [50]. Experimental evidence has demonstrated the anti-proliferative role of MBD3, acting as a protective mechanism against glioblastoma development. Clinical studies in glioma patients revealed that MBD3, in conjunction with 5hmC, was associated with improved progression-free survival (PFS) and overall survival (OS) [50].

### 3.2. Methyl Readers in Other Developmental Processes

Methyl CpG Binding Protein 2 (MeCP2) is crucial for fine-tuning gene transcription and regulating alternative splicing in response to neuronal stimulation [51]. MeCP2, along with splicing factor Y-box binding protein 1 (YB-1) and Ten-eleven translocation methylcytosine dioxygenase 1 (Tet1), bind to Brain Derived Neurotrophic Factor (BDNF)-associated chromatin and affect DNA methylation and splicing regulation [51]. 

MeCP2 generates two isoforms, MeCP2-E1 and MeCP2-E2, each with unique N-terminal domains and expression patterns in different brain regions. These isoforms exhibit distinct DNA interactions and dynamics, influencing gene regulation. Mutations in either isoform are associated with Rett syndrome, a severe neurological disorder [52,53]. MeCP2 interacts with RNA splicing modulators, and mutations causing Rett syndrome disrupt these interactions, leading to splicing deregulation [54]. MeCP2 also plays a role in learning-dependent post-transcriptional responses in the hippocampus, affecting splicing modalities and gene regulation [55].

Methyl-CpG Binding Domain Protein 1 (MBD1) is a methyl-CpG binding protein, with its major isoform, MBD1a, containing a CXXC domain that binds non-methylated CpG sites, enabling it to repress gene expression in a context-dependent manner regardless of DNA methylation status [56]. The related family member MBD5 has isoforms that display tissue-specific expression patterns, with full-length MBD5 predominant in the brain and alternatively spliced MBD5 mainly found in oocytes. Mutational analyses indicated that both the MBD and PWWP domains are necessary, but not sufficient, for MBD5 recruitment to methylated pericentric heterochromatin [57]. 

A rare genetic disorder, 2q23.1 Microdeletion Syndrome, is characterized by intellectual disability, motor delay, and distinctive craniofacial features due to haploinsufficiency or deletion of the *MBD5* gene. Studies have supported a role for MBD5 in neuronal development, linking genetic aspects to disease pathogenesis [58]. However, an individual with a similar phenotype had an intronic deletion in the 5’UTR of *MBD5* and showed normal mRNA expression, raising questions about the causative role in 2q23.1 deletion syndrome [59].

ASH1 Like 1 Histone Lysine Methyltransferase (ASH1L) has been implicated in autism spectrum disorders (ASD). Specifically, alternative splicing of Neurotrophic Receptor Tyrosine Kinase 2 (*NTRK2/TrkB*) mRNA in forebrain cortical neurons affected TrkB isoform expression and neuronal arborization [60]. 

Methyl-CpG Binding Domain Protein 2 (MBD2) isoforms have contrasting roles, with MBD2a promoting cell differentiation and MBD2c enhancing reprogramming to pluripotency. SFRS2, regulated by OCT4, influenced the alternative splicing of *MBD2* through miR-301-302 [61]. The related family member MBD4 is a DNA glycosylase that specifically targets G:T and G:U mismatches. An alternatively spliced variant of MBD4 gains methyl-binding capacity, enabling it to target mutagenic CpG regions by recognizing a novel cryptic splice site [62]. 

SET Domain Containing 2 Histone Lysine Methyltransferase 2 (SETD2) protein is a histone H3K36 methyltransferase that regulates alternative splicing. Genetic knockdown of *Setd2* enhanced the inclusion of PTB-dependent exons in *MRG15* (*MORF-related gene 15*) alternative splicing without affecting PTB-independent exons or constitutively spliced exons [63,64]. 

Ubiquitin Like with PHD and Ring Finger Domains 1 (UHRF1), is a multifunctional protein that links histone modifications and DNA methylation with histone ubiquitylation. Alternatively spliced UHRF1 featured a coupled TTD-PHD module with a unique linker, leading to enhanced H3K9me3-nucleosome ubiquitylation activity and distinct cellular localization [65].

CCCTC-binding factor (CTCF) is a versatile protein with zinc finger transcription factor properties that is pivotal for chromatin organization, genomic insulation, X-chromosome inactivation, and gene regulation. Alternatively spliced CTCF competed with canonical CTCF for genome binding, disrupting the interaction with cohesin and leading to alterations in CTCF-mediated chromatin looping [66].

EGR1 is a transcription factor that controls genes related to diverse biological functions, including growth, cell proliferation, and apoptosis [67]. Aberrant EGR1 expression has been linked to various disorders, including neurological diseases and cancer [68]. In a colon cancer study, a combination of erlotinib and sulindac drug treatment significantly reduced *Egr1* expression [69]. Alternatively spliced EGR1 lacks the N-terminal activation domain that is found in canonical EGR1, while the alternatively spliced form can enter the nucleus but is incapable of activating transcription [67]. More details on the effects of alternative splicing events on various methylation readers are compiled in Table 2.

## 4. Alternative Splicing of Phosphoryl Readers

### 4.1. Phosphoryl Readers in Cancer

BRCA1-associated RING domain protein 1 (BARD1) is essential for cellular functions such as DNA repair and cell cycle control. Multiple *BARD1* mRNA isoforms exist with various exon compositions due to alternative splicing [70]. Full-length *BARD1* mRNA encodes a protein with essential domains for protein-protein interactions and ubiquitin ligase activity. Specific BARD1 isoforms are linked to cancer; for example, BARD1β is associated with neuroblastoma susceptibility, while BARD1δ, an isoform with a deletion, is linked to tumor progression and genomic instability. Overexpression of BARD1 isoforms can have distinct, non-redundant effects on cell proliferation and cancer [70].

### 4.2. Phosphoryl Readers in Other Developmental Processes

The 14-3-3 proteins play a critical role as cellular scaffolds. Depletion of 14-3-3 significantly impacted alternative splicing during adipocyte differentiation [71]. 14-3-3 interactome comprised splicing factors like Heterogenous Ribonuclear Protein K (HNRPK), Heterogenous Ribonuclear Protein F (HNRPF), Splicing Factor Proline and Glutamine Rich (SFPQ), and DEAD Box Helicase 6 (DDX6), influencing the alternative splicing of key genes like Peroxisome Proliferator Activator Receptor Gamma (*Pparg*) and *Lipin1* implicated in adipogenesis [71]. More details on the effects of alternative splicing events on these phosphorylation readers are compiled in Table 3.

## 5. Alternative Splicing in Butyryl Readers

### Butyryl Readers in Cancer

Bromodomain containing protein 9 (BRD9), a principal component of the non-canonical BAF chromatin-remodeling complex, is influenced by splice factor mutations. For example, mutant SF3B1 (Splicing Factor 3B1) recognizes an atypical intronic branch point in *BRD9*, resulting in alternative splicing and the incorporation of an exon from an endogenous retroviral element. The concomitant degradation and loss of BRD9 protein expression disrupt non-canonical BAF, promoting melanoma [72]. 

YEATS domain containing 2 (YEATS2), a structural component within the Ada-two-A-containing (ATAC) complex, a highly conserved histone acetyltransferase (HAT) complex, downregulates multiple genes involved in spliceosome assembly and activity. Downregulated genes show enrichment in cancer-related pathways, indicating that YEATS2 is involved in regulating the proliferation and viability of different types of tumors by modulating the transcription of essential genes [73]. 

Double PHD Fingers 2 (DPF2), a key factor in BAF chromatin remodeling complexes, undergoes alternative splicing regulated by Polypyrimidine Tract Binding Protein 1 (PTBP1) during neuronal development, impacting chromatin organization, gene regulation, and stem cell differentiation [74]. DPF2 is also implicated in multiple cancers, including acute myelogenous leukemia [75] and cervical cancer [76]. More details on the effects of alternative splicing events on these butyryl readers are compiled in Table 4.

## 6. Alternative Splicing of RNA-Binding Reader Proteins

### 6.1. RNA-Binding Readers in Cancer

FUSE-binding protein (FBP) and its repressor, FBP-interacting repressor (FIR), are engaged in the regulation of c-Myc expression and cellular proliferation [77]. Overexpression of FIR in hepatocellular carcinoma (HCC) promotes tumor progression and dedifferentiation by stimulating FBP expression through Transcription Factor DP1/E2F Transcription Factor (TFDP1/E2F1). Multiple splice variants of FIR expressed in HCC contribute to tumor promotion [78]. Targeting FIR pharmacologically may offer therapeutic potential for HCC patients with elevated FIR expression. 

Musashi RNA binding protein 2 (MSI2), an RNA-binding protein, regulates alternative splicing and photoreceptor protein expression in mature photoreceptor cells, impacting light response and cell survival [79]. In TNBC, the MSI2a isoform is downregulated and is linked to poor survival. MSI2a acts as a tumor suppressor by stabilizing *TP53INP1* mRNA, reducing TNBC cell invasion, and inhibiting ERK1/2 activity [80]. MSI2 has diverse roles, from sensory perception in photoreceptor cells to suppressing tumor progression in TNBC by regulating gene expression and cell invasion.

In metastatic bladder cancer patients, reduced RNA binding motif X-linked (RBMX) levels are linked to a poor prognosis. RBMX acts as a tumor suppressor, inhibiting tumor cell growth, migration, and invasion. RBMX prevents the formation of a specific cancer-promoting PKM isoform (Pyruvate Kinase M2 or PKM2), reducing cancer aggressiveness and glycolysis [81].

### 6.2. RNA-Binding Readers as Modulators of Splicing

RNA-binding proteins recognize specific RNA sequences that impact alternative splicing. For example, HIV TAT Specific Factor 1 (TAT-SF1) binds to the SF3b1 subunit of the U2 snRNP spliceosome complex and regulates gene expression [82]. TAT-SF1 also plays a crucial role as a co-factor in HIV-1 transcription, and its knockdown reduces viral infectivity by affecting post-transcriptional processes, including altering the ratio of unspliced to spliced viral transcripts [83].

RNA Binding Motif Protein 15 (RBM15) regulates RNA N6-methyladenosine (m6A) methylation. RBM15 is methylated by PRMT1 (Protein Arginine Methyl Transferase 1), leading to ubiquitin-mediated degradation. RBM15 binds to intronic regions of essential genes in megakaryopoiesis, like TAL1 (TAL bHLH Transcription Factor 1), RUNX1 (RUNX Transcription Factor), GATA1 (GATA binding protein 1), and c-MPL (c-MPL proto-oncogene), influencing alternative splicing processes [84].

RNA Binding Motif 38 (RBM38) is a versatile RNA-binding protein involved in distinct aspects of RNA regulation, interacting with the p53 family, and impacting mRNA translation. In late erythroid differentiation, RBM38 modulates alternative splicing in the Erythrocyte Membrane Protein Band 4.1 (*EPB41*) gene [85]. RBM38 also binds to SMEK homolog 2 (*SMEK2*) RNA and reduces its transcription elongation defect [86]. Moreover, RBM38 plays a vital role in B19V pre-mRNA splicing, which is essential for viral replication and virion release. RBM38 contributes to various RNA-related processes, encompassing splicing control and viral RNA processing [87].

RBMX, also known as HNRNPG, has critical roles in different aspects of cellular function. RBMX influences alternative splicing of α-synuclein, a protein linked to Parkinson’s disease (PD). By regulating exon skipping in α-synuclein, RBMX is associated with disease pathogenesis, suggesting a therapeutic avenue for PD [88]. 

RBMX is also involved in X-linked Intellectual Differentiation (XLID) and neuronal differentiation. Depletion of RBMX activates the p53 pathway [89]. Methylation of RBMX by PRMT5 affects its interaction with Serine and Arginine rich Splicing Factor 1 (SRSF1), leading to alternative splicing of the MDM4 regulator of p53 (*MDM4*) and p53 pathway deregulation [89]. Thus, RBMX plays critical roles in neurodegenerative diseases, cancer, and neuronal development, making it a potential target for disease prognosis and therapeutic intervention. More details on the effects of alternative splicing events on various RNA binding readers are compiled in Table 5.

## 7. Therapeutic Modulation of Alternative Splicing in Cancer

As noted above, alternative splicing plays a significant role in cancer etiology, with implications for drug resistance and radiosensitivity [90]. Drug resistance in cancer cells can result from alternative splicing-mediated changes in drug uptake, molecular targets, DNA repair, and survival signals. Genes, including Androgen Receptor and Survivin-3b, exhibit alternative splicing-related drug resistance [91]. Alternative splicing can also influence radiosensitivity, notably through p73 modulation, highlighting its clinical importance in cancer treatment [92]. Understanding the role of alternative splicing in drug resistance could be crucial for developing more effective therapies, including specific alternatively spliced epigenetic reader proteins in precision medicine.

RNA-based treatments, including antisense oligonucleotides (ASOs) and small interfering RNAs (siRNAs), are tailored to target the alternatively spliced variants [92] of epigenetic reader proteins. For example, ASOs have been effectively used to modify the alternative splicing of *BRD9* in certain SF3B mutant cancers, leading to the restoration of BRD9 protein expression and a subsequent decrease in tumor size [92]. Another example involves a specialized ASO that targets transcripts containing exon 5 of the *WT1* gene, triggering cell death in HL60 leukemia cells. This approach is distinct from using an ASO that targets all four known alternatively spliced *WT1* transcripts simultaneously [93].

Splice-switching oligonucleotides (SSOs) alter pre-mRNA splicing via spliceosomes [94], resulting in the production of proteins with unique functionalities. For example, an SSO that binds to the intron-exon junction of MAP kinase-interacting serine/threonine-protein kinase 1 (*MKNK*) prevents the generation of the *MKNK2-2b* splice variant, thereby making cancer cells more responsive to drug treatment [94].

Another potential therapeutic strategy is the use of target-specific siRNAs, which are double-stranded RNA fragments selectively interacting with target mRNAs, leading to mRNA cleavage by Argonaut 2 [92] in the RNA-induced silencing complex. However, siRNA’s clinical utility is limited by potential off-target effects and the activation of innate immune responses, including Toll-like receptor activation [95].

Chemical inhibition of alternative splicing is a promising approach for treating cancers with splicing abnormalities. Natural compounds, including those from plants, microorganisms, and marine species, exhibit anticancer activity [96]. For example, SF3B Complex Inhibitors, derived from natural products, target the SF3B complex, disrupting its interaction with RNA [97,98]. This category includes drugs like spliceostatin, sudemycin, pladienolide, E7107, H3B-8800, and herboxidiene analogs [99]. Spliceostatin A and pladienolide have demonstrated anti-proliferative activity in HeLa cells, while sudemycin has elicited antitumor responses in lymphocytic leukemia. E7107 inhibits spliceosome assembly in solid tumors [99,100]. H3B-8800, targeting spliceosome-mutant epithelial and hematologic tumor cells, modulates the SF3b complex [101]. Herboxidiene, a potent spliceosome inhibitor, shows anti-tumor activity in multiple cancers [99].

To address concerns regarding the direct inhibition of core splicing activities, researchers have shifted their focus to accessory splicing factors [99]. Aryl sulfonamide molecules, such as indisulam, tasisulam, E7820, and chloroquinoxaline sulfonamides, function by connecting accessory splicing factors RNA Binding Motif 39 (RBM39) and RNA Binding Motif 23 (RBM23) with the E3 ubiquitin ligases Cullin 4A-RING/DDB1 and CUL4 associated factor 15 (CRL4/DCAF15), leading to proteasomal degradation and splicing defects [102].

Recent developments in small-molecule therapy have focused on targeting the formation and assembly of small nuclear ribonucleoproteins (snRNPs). This includes inhibiting protein arginine methyltransferases (PRMTs), a strategy showing promise in preclinical studies, particularly for leukemias with mutant alternative splicing factors. PRMT5 and type I PRMT antagonists, which inhibit both asymmetric and symmetric arginine dimethylation, are notable examples [103,104]. Type I PRMT inhibitors, such as GSK3368715, MS023, SGC707, TP-064, SGC6870, AMI-1, Allantodapsone, EZM2302, EPZ0220411, and MSO49, target various PRMTs and have shown nanomolar binding efficiency [103,104,105]. PRMT5 inhibitors, including EPZ-15666, GSK3326595, LLY-283, and JNJ-64619178, have been developed with enhanced selectivity and efficacy, with several, including JNJ-64619178, currently undergoing Phase I clinical trials [103,104,105]. Table 6 displays the chemical structures and targets of splicing inhibitors, highlighting their role in cancer therapy. Overall, exploring alternative splicing mechanisms offers promising therapeutic avenues for targeting cancer cells, as demonstrated in Figure 2. Continued research is needed in cancer development and other pathological conditions, as discussed in this review.

## 8. Discussion

Epigenetic reader proteins are involved in fundamental cellular processes such as DNA repair, chromatin remodeling, and transcriptional regulation and can fine-tune responses to changing environmental cues or developmental stages through alternative splicing. This adaptability may be particularly important during cellular differentiation, tissue development, and homeostasis, providing an Achilles heel for more effective treatment of cancer or other pathological conditions.

For example, in silico analyses were performed for some alternatively spliced proteins with reported antagonistic functions in cancer, including BRD4, KLF4, and UHRF2. BRD4-L and BRD4-S share an identical N-terminal region (residues 1–722), but BRD4-S is truncated at the C-terminal end. The latter contains two bromodomains (BD1 and BD2), an extraterminal domain (ET), a basic residue-enriched interaction domain (BID), and a C-terminal cluster of phosphorylation sites (CPS). BRD4-L additionally contains an intrinsically disordered region (IDR) that entails the N-terminal cluster of phosphorylation sites (NPS) and a P-TEFb interaction domain (PID) (Figure 3a). Full-length BRD4-L was retrieved from the AlphaFold database [106,107] via UniProt accession code Q5BJ26, whereas the short BRD4-S variant was accessed via UniProt accession code O60885-2 and modeled using the standalone version of Alphafold v.2.3.2 [108] (Figure 3a). In silico analyses of short vs. long BRD4 proteins revealed a notable structural disparity, with a root-mean-square deviation (r.m.s.d.) of 33.63 (Figure 3a). This correlated with the reported alteration in functionality between BRD4-L and BRD4-S, the former being involved in breast cancer suppression and the latter in breast cancer progression [24,25]. 

KLF4-L contains TAD, ZNF296, CZH2-1, 2, 3, and DNA-binding domains, whereas KLF4-S harbors only the N-terminal TAD region (Figure 3b). Comparative in silico analyses revealed a substantial structural difference, as expected, with an r.m.s.d. of 12.95 Å (Figure 3b). This observation corresponds with the differences in functionality between KLF4 isoforms: KLF4-L is associated with tumor suppression in breast tissues, and KLF4-S is designated as having an oncogenic role in breast cancer [46].

UHRF2-L and UHRF2-S share N- and C-terminal end segments, but UHRF2-S is truncated in the middle region. UHRF2-S harbors the ULD, PHD, and RING domains, whereas UHRF2-L additionally contains the SRA-YDG domain (Figure 3c). In silico analysis of short vs. long UHRF2 proteins indicated that the two structures differed markedly, with an r.m.s.d. of 4.63 Å (Figure 3c). This structural alignment coincides with the differences in functionality between UHRF2-L, known for its tumor suppressive properties, and UHRF2-S, which is associated with oncogenesis in breast and pancreatic tissues [49].

Most of the epigenetic readers discussed in this review directly undergo alternative splicing, and the corresponding splice variants are implicated in cellular processes including splicing modulation, chromatin remodeling, transcription, stem cell differentiation, protein-protein interactions, DNA damage responses, bone morphogenesis, spermatogenesis, neurodevelopment, heat stress, and disease pathogenesis, including cancer and neurodegenerative disorders (Figure 4). Some readers exhibit indirect regulation by mimicking alternative splicing patterns of various genes, thereby influencing histone modifications, cellular motility, megakaryopoiesis, erythroid differentiation, breast cancer stem cell differentiation, synaptic transmission, and lipogenesis (Figure 4). An improved mechanistic understanding of such alternatively spliced events will offer better insights into rational drug design and targeted precision medicine. In situations where most or all isoforms are oncogenic, broad-spectrum antagonists may be used to control tumor progression. Conversely, when only one splice variant is pro-tumorigenic while others have tumor-suppressing properties, selective inhibitors targeting the specific pro-tumorigenic isoform might be more appropriate. 

## 9. Conclusions

The intricate nature of alternative splicing regulation, with its temporal and tissue-specific nuances, presents formidable yet surmountable challenges for precision oncology. Advancements in genomics and transcriptomics, augmented by cutting-edge computational techniques, are poised to demystify the complexities of alternative splicing in disease pathogenesis, including cancer. Delving into the distinct functionalities of various splice forms of epigenetic readers emerges as a vital scientific pursuit. This endeavor not only holds the key to novel therapeutic interventions but also promises to significantly deepen our comprehension of epigenetic regulation in both health and disease. Future research should prioritize refining the identification and targeting of specific epigenetic reader splice variants for the development of personalized cancer therapies. Overcoming challenges related to specificity, delivery, and resistance mechanisms is essential for translating these findings into clinically impactful cancer treatments.

## Figures and Tables

**Figure 1 cancers-16-01104-f001:**
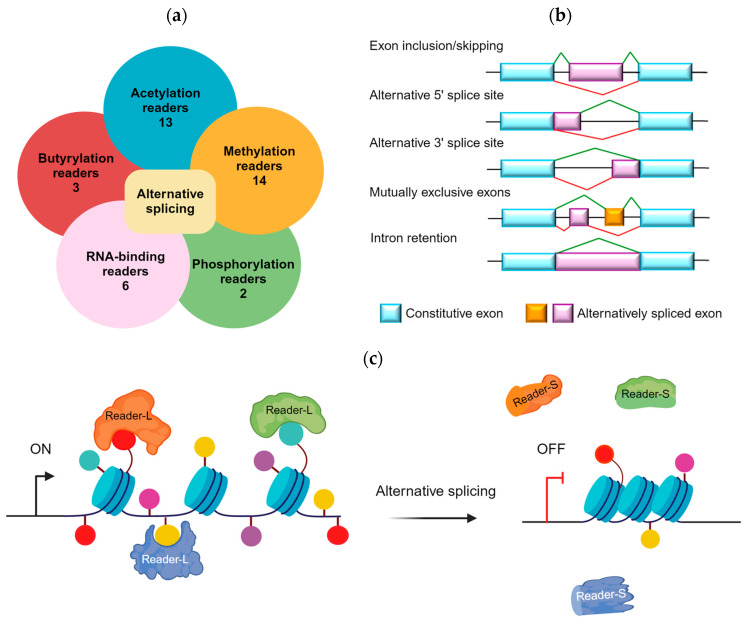
Overview of alternative splicing in epigenetic readers. (**a**) Distribution of epigenetic reader proteins by post-translational modification, indicating the number of distinct proteins involved in alternative splicing. (**b**) Schematic of alternative splicing mechanisms, highlighting exon and intron dynamics, with color-coding representing constitutive and alternatively spliced exons. (**c**) A model depicts gene regulation by epigenetic readers. In the ‘ON’ state, long-form readers (Reader-L) bind to chromatin, promoting gene expression. The ‘OFF’ state is mediated by the short-form readers (Reader-S) resulting from alternative splicing, which suppresses gene activity. This visualization emphasizes the importance of epigenetic reader length in gene regulation, a key factor in precision oncology strategies. Created with BioRender.com, accessed on 1 March 2024.

**Figure 2 cancers-16-01104-f002:**
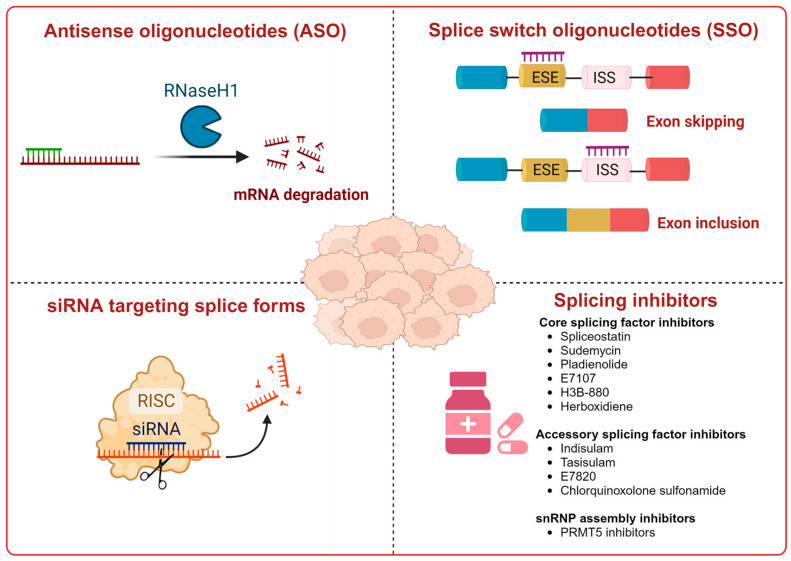
RNA-based therapeutics targeting splicing in cancer. This figure summarizes RNA therapeutics targeting alternative splicing in cancer. It includes ASOs that correct splicing errors, restore normal protein function, SSOs that repair transcripts, siRNA for selective mRNA degradation, SF3B Complex Inhibitors disrupting RNA-SF3B binding, and compounds leading to splicing factor degradation or snRNP biogenesis disruption. Created with BioRender.com, accessed on 2 February 2024.

**Figure 3 cancers-16-01104-f003:**
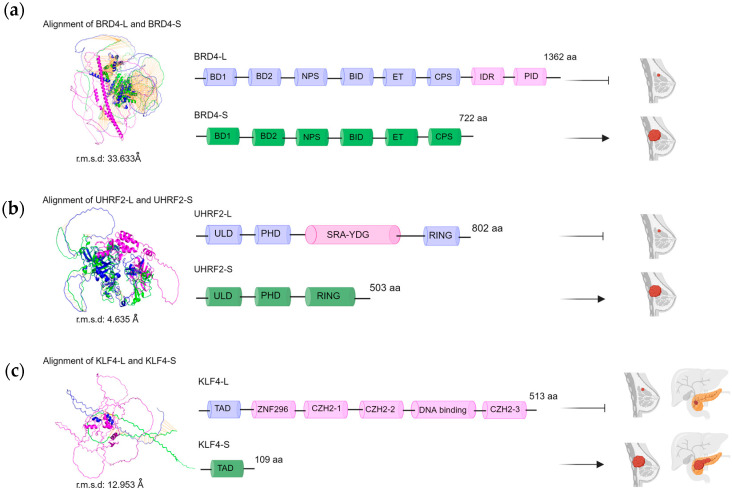
Comparative structural and functional profiles of BRD4, UHRF2, and KLF4 isoforms in cancer analysis of full-length and alternatively spliced protein isoforms: (**a**) BRD4 (O60885-1 and O60885-2), (**b**) UHRF2 (Q96PU4-1 and Q96PU4-2), and (**c**) KLF4 (O43474-3 and O43474-5), comparing their 3D structures (superimposition and r.m.s.d.), domain organization, and implications for breast and/or pancreatic cancer. Created with BioRender.com, accessed on 2 February 2024.

**Figure 4 cancers-16-01104-f004:**
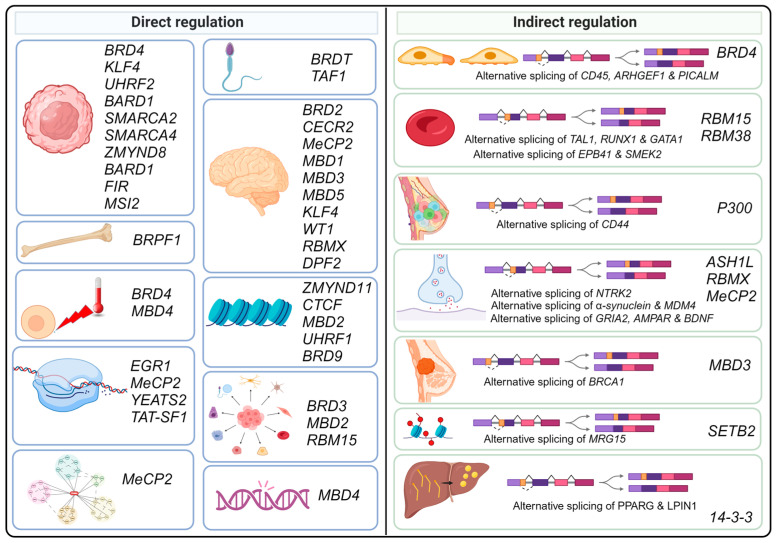
Epigenetic readers’ role in alternative splicing regulation. This figure presents the dual role of epigenetic readers in the regulation of alternative splicing, where they directly undergo alternative splicing affecting key biological processes like bone morphogenesis, neurodevelopment, and stress response, as well as indirectly altering splicing patterns in genes linked to cellular functions and diseases, including cancer. It highlights key players such as bromodomain proteins (BRD2, BRD4, BRDT), PHD finger-containing proteins, nucleosome remodeling factors, TAF1, zinc finger proteins, histone lysine methyltransferases, methyl-CpG binding proteins, and others, underscoring their significance in cellular mechanisms and disease pathology. Created with BioRender.com, accessed on 1 February 2024.

**Table 1 cancers-16-01104-t001:** Effects of alternative splicing on acetylation readers.

Acetylation Reader	Splicing Event	Functional Implication	References
BRD2	Inclusion of exon 2a; Exon 7 and intron inclusion in *IL17RC*; Intron retention in *DUSP2*	Neural development; Signal transduction modulation	[41,42]
BRD4	Intron retention; Fusion of exon 15 (*BRD4*) with exon 2 (*NUT*); Cassette exons in *CD45*, *Arhgef1*, and *Picalm*	Heat stress response; Oncogenic fusion; Signal transduction; Cell communication	[21,22,23]
BRD3	Exon skipping	Nuclear reprogramming	[26]
BRDT	Intron 6 retention	Spermatogenesis	[39]
BRPF1	Cassette exon 9	Osteoclastogenesis	[44]
CECR2	Exon 2–8 skipping	Neural development	[43]
NURF	Intron retention (partial exon 6)	Spermatocyte differentiation	[37]
P300	Exon skipping/inclusions in CD44	Signal transduction	[35]
SMARCA2	Alternative 3′ splice site in exon 1; Exon 2–5 skipping	Tumorigenesis	[27]
SMARCA4	Alternative TG splice site in intron 28; Extra exon between exons 26 and 27	Spermatogenesis	[29,30,31,32]
TAF1	Cassette exons 12a and 13a	Spermatogenesis	[40]
ZMYND8	Exon 22 inclusion	Breast cancer	[34]
ZMYND11	Intron retention	Chromatin regulation in pre-mRNA processing	[36]

Arhgef1, Rho Guanine Nucleotide Exchange Factor 1; BRD2, Bromodomain Containing Protein 2; BRD3, Bromodomain Containing Protein 3; BRD4, Bromodomain Containing Protein 4; BRDT, Bromodomain Testis Associated; BRPF1, Bromodomain and PHD Finger Containing Protein 1; CECR2, Cat Eye Syndrome Chromosome Region Candidate 2; CD44, Cluster of Differentiation 44; CD45, Cluster of Differentiation 45; DUSP2, Dual Specificity Phosphatase 2; IL17RC, Interleukin 17 Receptor C; NURF, Drosophila Nucleosome Remodeling Factor; NUT, Nuclear Protein in Testis; P300, Protein 300; Picalm, Phosphatidylinositol Binding Clathrin Assembly Protein; SMARCA2, SWI/SNF Related, Matrix Associated, Actin Dependent Regulator of Chromatin, Subfamily A, Member 2; SMARCA4, SWI/SNF Related, Matrix Associated, Actin Dependent Regulator of Chromatin, Subfamily A, Member 4; TAF1, TATA-box Binding Protein Associated Factor 1; ZMYND8, Zinc Finger MYND-type Containing 8; ZMYND11, Zinc Finger MYND-type Containing 11.

**Table 2 cancers-16-01104-t002:** Effects of alternative splicing in methyl reader proteins.

Methylation Reader	Splicing Event	Functional Implication	References
ASH1L	Exon 3–5 inclusion	Neuronal morphogenesis	[60]
CTCF	Exon 3–4 skipping	Chromatin regulation	[66]
EGR1	Exon 2 skipping	Transcriptional activation alteration	[67]
KLF4	Exon 3 deletion	Oncogenic activity in various cancers	[45,46]
MBD1	Exon 10 skipping, Alternative 3′ end	Transcriptional regulation	[56]
MBD2	Alternative 3′ end	Stem cell differentiation, Chromatin remodeler interaction	[61]
MBD3	Exon 9–11 skipping	Breast tumorigenesis	[50]
MBD4	Exon 3 skipping	Metabolic activity alteration	[62]
MBD5	Intron 9 retention, Exon 10–15 skipping, Intron 11 retention, Exon 12 and 14 skipping, Inherited intronic deletion in 5′-UTR	Neurodevelopment, Cellular localization, and Abundance changes	[57,58,59]
MeCP2	Exon 2 skipping, Intron retention, Flop exon inclusion	Neurodevelopment, Cognition, and Synaptic transmission	[51,52,53,54,55]
SETD2	Exon inclusion	Chromatin regulation	[63,64]
UHRF1	Intron retention	Chromatin regulation	[65]
UHRF2	Exon 10 inclusion	Breast tumor suppression	[49]
WT1	Intron retention	Tumorigenesis	[47]

ASH1L, ASH1 Like Histone Lysine Methyltransferase; CTCF, CCCTC-binding Factor; EGR1, Early Growth Response 1; KLF4, KLF Transcription Factor 4; MBD1, Methyl-CpG Binding Domain Protein 1; MBD2, Methyl-CpG Binding Domain Protein 2; MBD3, Methyl-CpG Binding Domain Protein 3; MBD4, Methyl-CpG Binding Domain Protein 4; MBD5, Methyl-CpG Binding Domain Protein 5; MeCP2, Methyl-CpG Binding Protein 2; SETD2, SET Domain Containing 2, Histone Lysine Methyltransferase; UHRF1, Ubiquitin Like with PHD and Ring Finger Domains 1; UHRF2, Ubiquitin Like with PHD and Ring Finger Domains 2; UTR, Untranslated Region; WT1, WT1 Transcription Factor.

**Table 3 cancers-16-01104-t003:** Influence of alternative splicing on phosphorylation reader function.

Phosphoryl Reader	Splicing Event	Functional Implication	References
BARD1	Exon 2–6 skipping	Tumor suppression	[70]
YWHAQ	Exon 3 skipping	Lipogenesis	[71]

BARD1—BRCA1 Associated RING Domain 1; YWHAQ—Tyrosine 3-monooxygenase/tryptophan 5-monooxygenase activation protein, theta polypeptide (14-3-3).

**Table 4 cancers-16-01104-t004:** Effects of alternative splicing on butyrylation reader function.

Butyryl Reader	Splicing Event	Functional Implication	References
BRD9	SF3B1 mutation-induced exonization of introns	Impairment of non-canonical BAF complex activity	[72]
DPF2	Exon 7 skipping	Neural cell differentiation	[74]
YEATS2	Exon inclusion	Downregulation of splicing genes	[73]

BAF, BRG1/BRM-Associated Factor; BRD9, Bromodomain Containing 9; DPF2, Double PHD Fingers 2; SF3B1, Splicing Factor 3b Subunit 1; YEATS2, YEATS Domain Containing 2.

**Table 5 cancers-16-01104-t005:** Alternative splicing in RNA binding readers.

RNA Reader	Splicing Event	Biological Implication	References
FIR	Exon-2 skipping	Oncogenesis in liver cancer	[78]
MSI2	Promotion of photoreceptor-specific alternate exons, Promotion of TP53INP1 mRNA stability	Photoreceptor survival, cancer cell invasion	[79,80]
RBM15	Interactions with intronic regions of pre-mRNA	Gene expression in blood cell development	[84]
RBM38	Splicing activation, ISE-2 interaction	Hematopoietic splicing regulation	[85,86,87]
RBMX	Intron-4 interaction, Disruption of hnRNPA1 and PKM interaction, Exon-6 exclusion in MDM4	Neurological disease modulation, splicing regulation	[81,88,89]
TAT-SF1	Intron inclusion	Spliceosome function, viral RNA processing	[82,83]

FUSE Interacting Repressor; hnRNPA1, Heterogeneous Nuclear Ribonucleoprotein A1; ISE-2, Intronic Splicing Enhancer 2; MDM4, Mouse Double Minute 4 Homolog; MSI2, Musashi RNA Binding Protein 2; PKM, Pyruvate Kinase M; RBM15, RNA Binding Motif Protein 15; RBM38, RNA Binding Motif Protein 38; RBMX, RNA Binding Motif Protein X-Linked; TAT-SF1, Tat-Specific Factor 1; TP53INP1, Tumor Protein P53 Inducible Nuclear Protein 1.

**Table 6 cancers-16-01104-t006:** Chemical structures and biological targets of alternative splicing inhibitors.

Drug Name	Chemical Structure	Target(s)
Spliceostatin	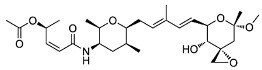	Inhibits the SF3B1 complex, affecting spliceosome function
Sudemycin	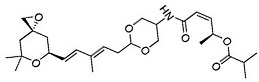	Same as Above
Pladienolide	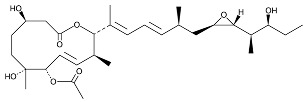	Same as Above
H3B-8800	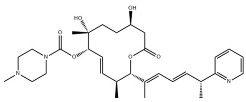	Same as Above
Herboxidiene	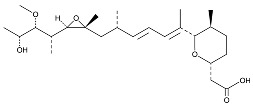	Same as Above
E7107	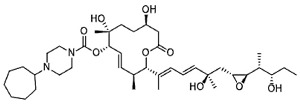	Disrupts spliceosome assembly by targeting spliceosome-associated protein-130
Indisulam	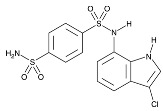	Targets RNA Binding Motif 39 (RBN39) for degradation
Tasisulam	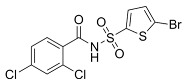	Apoptosis induction via the intrinsic pathway in cancer
E7820	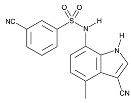	Interferes with cell adhesion and metastasis by targeting Integrin alpha2
Chloroquinoxaline sulfonamides	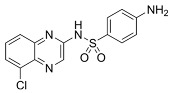	Inhibits Topoisomerase II alpha/beta, affecting DNA replication/cell division
GSK3368715	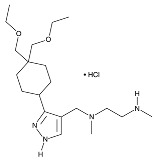	Inhibits protein arginine methyltransferases (PRMTs), affecting RNA splicing
MS023	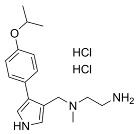	Same as Above
SGC707	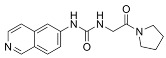	Same as Above
TP-064	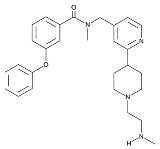	Same as Above
SGC6870	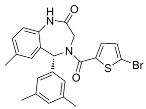	Same as Above
AMI-1	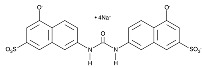	Same as Above
EZM2302	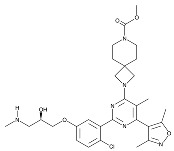	Same as Above
EPZ0220411	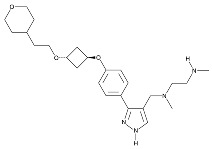	Same as Above
MS049	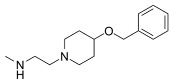	Same as Above
Allantodapsone	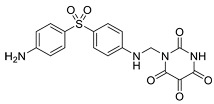	Targets bacterial adhesins like *Staphylococcus aureus* ClfA and ClfB
EPZ-15666	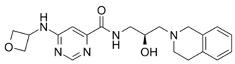	Specifically inhibits PRMT5, affecting RNA splicing
GSK3326595	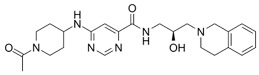	Same as Above
LLY-283	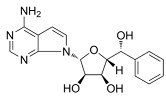	Same as Above
JNJ-64619178	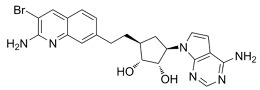	Same as Above

## Data Availability

The data can be shared on request.

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
