# Peer review of "A–Z of Epigenetic Readers: Targeting Alternative Splicing and Histone Modification Variants in Cancer"

_cancers, 2024, doi:10.3390/cancers16061104_

Round 1
Reviewer 1 Report
Comments and Suggestions for Authors
This is a well written review article about epigenetic readers and alternative splicing. This article is beneficial for both basic scientists and clinicians.
I have some suggestions to improve it.
1) It would be nicer if the authors could add the model Figures of alternative splicing regulation for some epigenetic readers. For instance, binding to splicing regulators or histones etc.
2) It would be wonderful if the authors could include Figures for the inhibitors to show their chemical structures and the names of binding targets.
Author Response
In response to Reviewer 1's suggestions, we have incorporated a new model figure (Fig 1c) to illustrate gene regulation by epigenetic readers and a comprehensive table (Table 6) detailing the chemical structures and targets of splicing inhibitors. These additions enrich our review by providing a clearer visual representation of the intricate relationship between epigenetic readers and alternative splicing in cancer.
Reviewer 2 Report
Comments and Suggestions for Authors
This manuscript reviews how splice variants present in histone modification readers affect the cell biology of cancers. The authors discuss readers of well studied histone modifications such as acetylation and methylation and some less well-known modifications. The review covers the important points. My only issue is that i think that the title is misleading. In reading the title i thought that the review would be discussing how epigenetics contributes to the choice of splice sites. However, mechanisms are not discussed at all. I think it would be better to have a title that more accurately represents the focus of the manuscript.
Author Response
Acknowledging Reviewer 2's concern regarding the initial title's potential for misinterpretation, we have revised it to reflect the manuscript’s content and focus more accurately, as suggested. The revised title is "A-Z of Epigenetic Readers: Targeting Alternative Splicing and Histone Modification Variants in Cancer"
Reviewer 3 Report
Comments and Suggestions for Authors
Strengths:
1) Thoroughly covers the landscape of epigenetic readers affected by alternative splicing across various cancers.
2) Highlights the potential for targeted therapies in precision oncology.
3) Offers detailed insights into the mechanisms by which alternative splicing influences epigenetic reader functionality.
Weaknesses:
1) Could benefit from a more explicit discussion on future research directions and the translational potential of these findings.
The authors are requested to elaborate on how these findings could be translated into clinical practice, including potential challenges.
2) In addition, the authors could incorporate case studies or real-world examples where alternative splicing of epigenetic readers has been targeted in therapy.
Author Response
Acknowledging Reviewer 3's feedback, we clarify that the sections “Therapeutic Modulation of Alternative Splicing in Cancer” and “Discussion” already delve into the translational potential and real-world applications of our findings, including future research directions and challenges in clinical translation. We've added a concise note to our conclusions to reiterate the significance of specificity, delivery, and overcoming resistance in developing personalized cancer therapies, directly addressing the points raised.
Round 2
Reviewer 3 Report
Comments and Suggestions for Authors
The authors have responded to the reviewer queries.